# The Gene and Protein Expression of the Main Components of the Lipolytic System in Human Myocardium and Heart Perivascular Adipose Tissue. Effect of Coronary Atherosclerosis

**DOI:** 10.3390/ijms21030737

**Published:** 2020-01-22

**Authors:** Małgorzata Knapp, Jan Górski, Janina Lewkowicz, Anna Lisowska, Monika Gil, Beata Wójcik, Tomasz Hirnle, Adrian Chabowski, Agnieszka Mikłosz

**Affiliations:** 1Department of Cardiology, Medical University of Bialystok, 15-089 Bialystok, Poland; malgo33@interia.pl (M.K.); anlila@poczta.onet.pl (A.L.); moniiag@gmail.com (M.G.); 2Department of Medical Sciences, Lomza State University of Applied Sciences, 18-400 Lomza, Poland; gorski@umb.edu.pl; 3Department of Cardiosurgery, Medical University of Bialystok, 15-089 Bialystok, Poland; janka.lewkowicz@wp.pl (J.L.); hirnlet@wp.pl (T.H.); 4Department of Physiology, Medical University of Bialystok, 15-089 Bialystok, Poland; wojcikb@umb.edu.pl (B.W.); adrian@umb.edu.pl (A.C.)

**Keywords:** perivascular adipose tissue, multivessel coronary artery disease, adipose triglyceride lipase, comparative gene identification-58, G0/G1 switch protein 2, hormone-sensitive lipase

## Abstract

The aim of our study was to examine the regulation of triacylglycerols (TG) metabolism in myocardium and heart perivascular adipose tissue in coronary atherosclerosis. Adipose triglyceride lipase (ATGL) is the major TG-hydrolase. The enzyme is activated by a protein called comparative gene identification 58 (CGI-58) and inhibited by a protein called G0/G1 switch protein 2 (G0S2). Samples of the right atrial appendage and perivascular adipose tissue were obtained from two groups of patients: 1—with multivessel coronary artery disease qualified for coronary artery bypass grafting (CAD), 2—patients with no atherosclerosis qualified for a valve replacement (NCAD). The mRNA and protein analysis of ATGL, HSL, CGI-58, G0S2, FABP4, FAT/CD36, LPL, β-HAD, CS, COX4/1, FAS, SREBP-1c, GPAT1, COX-2, 15-LO, and NFκβ were determined by using real-time PCR and Western Blot. The level of lipids (i.e., TG, diacylglycerol (DG), and FFA) was examined by GLC. We demonstrated that in myocardium coronary atherosclerosis increases only the transcript level of G0S2 and FABP4. Most importantly, ATGL, β-HAD, and COX4/1 protein expression was reduced and it was accompanied by over double the elevation in TG content in the CAD group. The fatty acid synthesis and their cellular uptake were stable in the myocardium of patients with CAD. Additionally, the expression of proteins contributing to inflammation was increased in the myocardium of patients with coronary stenosis. Finally, in the perivascular adipose tissue, the mRNA of G0S2 was elevated, whereas the protein content of FABP-4 was increased and for COX4/1 diminished. These data suggest that a reduction in ATGL protein expression leads to myocardial steatosis in patients with CAD.

## 1. Introduction

It is well known that cardiomyocytes contain triacylglycerols (TG). They are localized in lipid droplets [1,2,3]. An excess of intracellular free fatty acids (FFA) is deleterious to the cells since it can result in lipotoxicity. However, much of the plasma FFA entering cardiomyocytes is incorporated into TG moieties before being further utilized, and the process prevents the accumulation of FFA in the cell. Interestingly, it was also shown that the endogenous TG provides a substantial amount of FFA, which are preferentially oxidized by the myocardium [4,5,6,7,8]. Proton magnetic resonance spectroscopy technique is allowed to measure myocardial TG content in humans in a noninvasive way in different states of health. For instance, Szczepaniak et al. [9], using the method, have reported the presence of TG in the myocardium of very lean, overweight, and obese individuals. Furthermore, an elevation of TG content in the heart has also been demonstrated in type 2 diabetes, metabolic syndrome [10,11,12], and in aging [13].

Originally it was believed that hormone-sensitive lipase (HSL) was the major, if not the only, lipase responsible for TG hydrolysis. Currently, it is well established that the adipose triglyceride lipase (ATGL) is a predominant, rate-limiting lipolytic enzyme responsible for the hydrolysis of the first ester bond of the TG moiety. As a result, diacylglycerol (DG) and FFA are released. Subsequently, diacylglycerols are further hydrolyzed by HSL to monoacylglycerol (MG) and free fatty acid. Eventually, monoacylglycerol is hydrolyzed to glycerol and a fatty acid by the enzyme monoacylglycerol lipase [14,15]. Considerable evidence has suggested that ATGL interacts with lipid droplet protein, called comparative gene identification 58 (CGI-58 or ABHD5). Recent studies have revealed that after hormonal stimulation, CGI-58 dissociates from perilipin A, interacts with ATGL, and activates lipolysis [16]. On the other hand, the enzyme activity of ATGL may be inhibited by the G0/G1 switch gene 2 (G0S2). It has been shown that G0S2 binds directly to the ATGL and attenuates ATGL-mediated lipolysis via inhibiting the activity of the enzyme [17,18]. Moreover, the process of lipolysis is facilitated by fatty acid-binding protein 4 (FABP-4), which binds the intracellular free fatty acids and traffics them to different organelles [19,20,21,22]. Undoubtedly, the important role of ATGL in lipolysis became evident from observations in ATGL-deficient mice. The absence of ATGL activity results in steatosis of the myocardium and heavy cardiac insufficiency [23]. Moreover, mutations in the gene encoding ATGL were also reported in humans. As a result, neutral lipid storage disease with myopathy (NLSDM) and different degrees of cardiomyopathy developed [24,25,26,27], thus confirming the key role of ATGL in maintaining the proper content of the myocardial TG.

Compared with a number of reports regarding the association of visceral fat with atherosclerosis, much less attention has been paid to the role of perivascular adipose tissue (PVAT) in the development of cardiovascular diseases [28]. This seems to be quite odd given the fact that the PVAT surrounds the outer layer of coronary artery vessels and secretes a number of bioactive adipokines, cytokines, and chemokines acting directly on the vascular wall. For these reasons, it has recently been proposed that the localization of PVAT is a risk factor for developing coronary artery disease (CAD) [29,30,31,32]. Unexpectedly, there are no studies evaluating the expression of the principal components of the TG-lipolytic system in the human perivascular adipose tissue. Jaffer et al. showed that in epicardial adipose tissue (EAT), the mRNA expression of ATGL, HSL, and CGI-58 in patients with coronary artery disease was similar to their expression in patients with aortic/mitral valve disease but without arteriosclerosis [33]. However, it should be emphasized that EAT and PVAT of coronary arteries have different phenotypes and unique properties, which distinguish them from each other. On the other hand, the presence of a common microcirculation network and direct anatomical proximity enables one to influence each other and also reach cardiomyocytes and modulate their function. Therefore, we examined both myocardial and perivascular adipose tissue expression of ATGL, CGI-58, G0S2, HSL, and FABP4 at the transcript (mRNA) and protein levels in patients with multivessel coronary artery disease and compared them with control patients without coronary atherosclerosis. Because the heart is a metabolically flexible organ with alterations in preferred substrate, this study have also explored compounds involved in fatty acid oxidation (beta-hydroxyacyl CoA dehydrogenase—β-HAD), tricarboxylic acid cycle (citrate synthase—CS), and mitochondrial electron transport chain (cytochrome c oxidase subunit IV isoform 1—COX4/1). We hypothesized that the observed increased TG level in human myocardium of patients with coronary artery disease is due to reduced ATGL activity, not to increased fatty acids de novo synthesis. To test this hypothesis, we evaluated the expression of fatty acid synthase (FASN) and SREBP-1c, a transcription factor responsible for regulating the genes required for de novo lipogenesis at the protein level. In addition, the expression of glycerol-3-phosphate acyltransferase 1 (GPAT1), an essential protein engaged in TG synthesis, was determined. Given the fact that the size of the myocardial TG pool is also determined by the facilitated FAs transport across the plasma membrane and the already established role of the FAT/CD36 as the major cardiac fatty acid transporter translocase, we measured its expression both at the mRNA and protein level. Additionally, the myocardial expression of lipoprotein lipase (LPL) and the level of lipids in the particular fraction i.e., TG, DG, and FFA, were also estimated. Finally, we measured the expression of proteins contributing to inflammation i.e., cyclooxygenase-2 (COX-2), nuclear factor kappa B (NFκβ) and lipoxygenase (15-LO) in human myocardium.

## 2. Results

### 2.1. Patient Clinical Characteristics

A total of 42 patients with multivessel coronary artery disease qualified for coronary artery bypass grafting (a study group, 40 males and 2 females, mean age 63.6 ± 8.3) and 11 patients without atherosclerosis, qualified for mitral or aortic valve replacement (a control group, 5 males and 6 females, mean age 61.1 ± 8.3) were enrolled in the study. The clinical data are summarized in Table 1. In the CAD group, 34 patients (80.9%) had hypertension and 17 (40.4%) had a history of myocardial infarction. The platelet count and alanine aminotransferase activity were significantly higher in the study group. Regarding medication, 40 subjects (95.2%) with CAD were treated with an angiotensin receptor blocker, 41 (97.6%) persons with statins, and 42 patients with aspirin. All the subjects who participated in the study were without diabetes mellitus.

### 2.2. Myocardial and Perivascular Adipose Tissue Expression of ATGL, CGI-58, G0S2, and HSL at the Transcript (mRNA) and Protein Levels in CAD and NCAD Patients

#### 2.2.1. Myocardium

In the human myocardium, the protein expression of ATGL was significantly lower in CAD patients as compared to the control group (−20%, *p* < 0.05, Figure 1A). Additionally, a trend toward a decrease in the expression of ATGL at the transcript (mRNA) level was also observed in the studied group (−28.1%, *p* = 0.11, Figure 2A). On the opposite, coronary atherosclerosis substantially elevated the mRNA level of G0S2 (+102.3%, *p* < 0.05, Figure 2C), however, it did not change its protein content. Finally, there were no significant differences in the expression of CGI-58 (an activator of ATGL) and HSL at both the mRNA and protein levels in the CAD patients.

#### 2.2.2. Perivascular Adipose Tissue (PVAT)

The expression of G0S2, an inhibitor of ATGL, was increased at the mRNA level in CAD patients (+50.5%, *p* < 0.05, Figure 2C), however, there were no significant alterations in its protein content (+22.7%, *p* < 0.05, Figure 1C). Correspondingly, there were no significant differences in both mRNA and protein expression of ATGL, CGI-58, and HSL between the studies groups in the PVAT (Figure 1 and Figure 2A,B,D).

### 2.3. Compounds Involved in Fatty Acid Metabolism i.e., β-HAD, CS, COX4/1, FAS, SREBP-1c, GPAT1, FAT/CD36, LPL, and FABP4 at the Transcript (mRNA) and Protein Levels in Myocardium and Perivascular Adipose Tissue of CAD and NCAD Patients

#### 2.3.1. Myocardium

The expression of β-HAD, which catalyzes the third step of beta-oxidation, was significantly reduced at both the mRNA and protein levels (−44% and −29%, *p* <0.05, Figure 3A and Figure 4A). Furthermore, the patients from the CAD group exhibited a lower protein expression of cytochrome c oxidase (complex IV) (−24%, *p* <0.05, Figure 4C). As illustrated in Figure 5B, atherosclerosis caused significant increases in the FABP4 mRNA level (+94.4%, *p* <0.05) in human myocardium. Additionally, neither the mRNA nor the protein levels of CS, FAT/CD36, or LPL as well as FAS, SREBP-1c and GPAT1 changed significantly in subjects with multivessel coronary artery disease (Figure 3B, Figure 4B, Figure 5A,C,D, Figure 6A and Figure 7A–C).

#### 2.3.2. Perivascular Adipose Tissue (PVAT)

Coronary atherosclerosis resulted in a 50.8% increase in FABP4 mRNA and a 19.3% increase in its protein content, although the statistical significance was achieved solely in the case of protein (Figure 5B and Figure 6B). The mRNA expression of the other compounds in the perivascular adipose tissue of the CAD group did not differ from the respective value in the NCAD group (*p* > 0.05, Figure 3A,B,C and Figure 5A,C). Regarding the protein level, there was a statistically significant decrease solely in the case of COX4/1 (−15%, *p* < 0.05, Figure 4C), whereas the protein expression of other compounds remained unchanged (*p* > 0.05, Figure 4A,B, Figure 6A and Figure 7C).

### 2.4. Myocardial Expression of Proteins Contributing to Inflammation, i.e., COX-2, NFκβ and 15-LO at the Protein Level of CAD and NCAD Patients

We observed that the expression of COX-2 in CAD patients was significantly increased (+50%, *p* < 0.05, Figure 8A) as compared with control patients. Moreover, despite not reaching the statistical significance level, it appears that atherosclerosis tended to increase NFκβ and 15-LO proteins expression (+23%, and +27%, *p* >0.05, respectively, Figure 8B,C).

### 2.5. Myocardial Lipid Content and Fatty Acid Composition in CAD and NCAD Patients

The total fatty acid content of triacylglycerols (TG) was significantly greater in CAD subjects (+121%, *p* <0.05, Figure 9C) in comparison with NCAD subjects. This was accompanied by a significant increment in the pool of the saturated (SAT), i.e., myristic (+114%), palmitic (+118%), stearic (+74%), arachidic (+74%), behenic (+217%), and lignoceric (+160%), and also unsaturated (UNSAT) i.e., palmitooleic (+162%), oleic (+124%), linoleic (+103%), arachidonic (+132%), eicosapentaenoic (+584%), and docosahexaenoic (+159%) fatty acids (Table 2). Although the total content of diacylglycerols (DG) reminded unchanged, we noticed an increase of linoleic (+33%, *p* <0.05, Table 3) and lignoceric (+158%, *p* <0.05, Table 3) fatty acids in the studied group (Figure 9B). Moreover, in comparison with the NCAD individuals, neither the total content of FFA nor the saturated or unsaturated, with the exception of eicosapentaenoic acid (+76%, *p* < 0.05, Table 4) fatty acids significantly changed in coronary atherosclerosis group (Figure 9A).

### 2.6. Correlation of ATGL Expression and TG Content in Human Myocardium

In both the studied groups (NCAD and CAD) there were negative correlations between ATGL expression and TG content in human myocardium (*p* <0.05, *r* = −0,6686 and r = −0,4867, respectively, Figure 10A,B).

## 3. Discussion

This study showed, for the first time, the presence of the principal components of the triacylglycerol lipolytic complex in the myocardium and perivascular adipose tissue in patients with coronary artery disease undergoing a coronary artery bypass. Herein, we found that severe coronary atherosclerosis affects both the mRNA and protein expression—the chief components of the above-mentioned complex. One of the most remarkable findings of this study is the reduction in the ATGL protein expression in the human myocardium of the CAD patients.

As already mentioned in the introduction, ATGL catalyzes the first and rate-limiting step of triacylglycerol hydrolysis and exhibits a 10-fold higher substrate specificity for TG as for DG. Real-time PCR analysis revealed that cardiac muscle has about 25% of the ATGL mRNA level found in white adipose tissue, which expressed the highest levels of mRNA and enzyme activity [34]. This study has one limitation because we did not measure the activity of the enzyme due to the lack of muscle samples. However, it is a good reason to believe that the reduction in the enzyme’s protein content would lead to a reduction in its lipolytic action. In line with this assumption are the data showing a 121% elevation in the content of triacylglycerols in the myocardium. Evidence suggests that excessive myocardial neutral fat deposition could contribute to the development of cardiovascular disease [13,35]. However, from a practical point of view, the direct assessment of myocardial TG content in humans is limited by the difficulty of obtaining samples of the tissue. For that reason, the proton magnetic resonance spectroscopy (1H-MRS) is used for monitoring TG content in human myocardial tissue. Recently, it was shown that myocardial TG content was elevated in patients with ischemic CAD in relation to patients with nonischemic CAD. The degree of ischemia was evaluated with the PET/CT technique [36]. In the present study, we did not measure a degree of ischemia but all the participants with CAD were in a stable condition. Nonetheless, our data obtained by direct measurement of the triacylglycerol content in the myocardium of the patients with multivessel coronary artery disease are in line with the above-mentioned report [36].

It has been demonstrated that the activity of ATGL has dual regulation, it is greatly enhanced by comparative gene identification-58 (CGI-58) [16] and inhibited by the G0/G1 switch gene-2 protein (G0S2) [17,18]. In the presence of CGI-58, the human ATGL activity is increased approximately 5-fold. Mutations in the human CGI-58 gene cause neutral lipid storage disease (NLSD) or Chanarin Dorfman Syndrome in humans, a genetic disease characterized by systemic TG accumulation in multiple tissues [3]. Lass et al. [16] have recently confirmed that the silencing of CGI-58 expression in 3T3-L1 adipocytes decreased the lipolytic mobilization of TG. These data emphasized that CGI-58 is physiologically important for the cellular metabolism of neutral lipids. The data presented herein demonstrate that the myocardial expression of lipase coactivator CGI-58 at mRNA and protein levels is not affected by atherosclerosis. Yang et al. reported that the G0S2, an inhibitor of ATGL, binds directly to ATGL and is capable of decreasing ATGL-mediated lipolysis via inhibiting its TG hydrolase activity [18]. In addition, in our study, we observed that the G0S2 mRNA level was higher in the myocardium of patients with CAD. It is postulated that the level of ATGL lipase inhibitor G0S2 is indicative of the lipolytic potential of the tissue. Our results might support decreased rates of lipolysis since we have observed an elevation in TG level in heart muscle. On the other hand, the protein expression of hormone-sensitive lipase, which is the rate-limiting enzyme in DG, but not TG hydrolysis, and diacylglycerol content were unchanged in the coronary atherosclerosis state. Given these results, it is possible that the second step of triacylglycerol lipolysis, i.e., hydrolysis of diacylglycerol, is not affected by coronary artery disease.

It is well known that fatty acids are the principal energy fuel of the heart. Increased TG deposition in the myocardium can result from increased fatty acid (FA) uptake or decreased FA oxidation [37]. Recently cardiomyocytes TG depot has been identified as a dynamic source of LCFA utilized mainly for mitochondrial energy production via beta-oxidation and the citric acid cycle [6]. Based on the fact that lipids metabolism is dysregulated in the failing heart, we have taken into account other metabolic routes of FA/TG that can be responsible for TG elevation in the myocardium of patients with multivessel atherosclerosis. Therefore, we examined mRNA and protein expression of the representative compounds engaged in β-oxidation (β-HAD), citric acid cycle (CS), and oxidative phosphorylation (COX4/1). We observed a significant decrease in the case of β-HAD (at both mRNA and protein levels) and protein expression of COX4/1 in the myocardium of CAD patients. Considering the constant mRNA/protein expression of FAT/CD36 and LPL in our study, it is unlikely that the increase in the TG level originates from an unchanged cardiac FA influx. The cardiac muscle is not able to autonomously synthesize fatty acids but rather relies on the exogenous supply from plasma [38] as is confirmed by low expression of proteins involved in de novo fatty acid synthesis (i.e., FASN). In line with that notion, we measured the expression of fatty acid synthase (FASN), which catalyzes the synthesis of long-chain fatty acids from acetyl-CoA and malonyl-CoA, SREBP-1c, a transcription factor responsible for regulating the genes required for de novo lipogenesis and GPAT1, a rate-limiting enzyme in TG synthesis. The expression of all the above-mentioned proteins was stable in the human myocardium. Therefore, it seems to us that an increase in the TG content probably comes from reduced ATGL activity, even though this change was not very high. To confirm our assumptions we have correlated the expression of ATGL and TG levels in human myocardium. In addition, we have observed statistically significant negative correlations between ATGL expression and TG content. Taken altogether, the presented results indicate that an increase in the myocardial TG content is a consequence of ATGL protein expression decrement. Taken altogether, the unchanged de novo fatty acid synthesis and stable fatty acid uptake indicate that the increased TG level in human myocardium of CAD patients is due to reduced ATGL activity.

Additionally, the diminished expression of enzymes involved in TG oxidation, i.e., β-HAD, COX4/1, also might contribute to the increase of the TG pool in the myocardium in coronary atherosclerosis.

Chronic inflammation is a typical hallmark of atherosclerosis, which may be induced and disturbed by lipid accumulation [39,40]. This statement is consistent with our results showing that the level of the inflammatory marker plasma C-reactive protein and the expression of proteins contributing to inflammation like COX-2, NFκβ, and 15-LO were increased in the myocardium of patients with coronary stenosis. Similarly to the increased COX-2 expression, CAD patients exhibit a higher level of arachidonic acid in the TG fraction, the acid being a substrate for prostaglandin synthesis. Furthermore, we observed that atherosclerosis could activate the NFκβ signaling pathway and elevate the expression of 15-LO (lipoxygenase), a protein that deoxygenates unsaturated fatty acids, thus acting in physiological membrane remodeling and the pathogenesis of atherosclerosis [41].

Nowadays, it is well known that human perivascular adipose tissue is an active fat depot that may promote the development of atherosclerotic coronary artery disease [28,30,31,32]. However, transcript and protein expression of the chief compounds regulating the triacylglycerol-lipolytic complex in the perivascular adipose tissue has never been previously explored in patients with coronary artery disease. Here, we found them unchanged as compared to the respective control patients with the exception of FABP4, which was significantly elevated at the protein level. It has recently been demonstrated that FABP4 is present both on coronary endothelium [42] and in cardiomyocytes [43] and significantly contributes to the induction and progression of coronary atherosclerosis [44]. Interestingly, the perivascular adipose tissue is not separated by any fibrous fascial layer from the myocardium and the two tissues are fed by the same coronary vessels [29,30,31,32]. Consequently, the FABP4 elevation in the perivascular adipose tissue would facilitate its movement into the adjacent coronary arteries. Eventually, it could contribute to the development of atherosclerosis of the arteries. However, in our study, atherosclerosis did not change either the myocardial content of FABP4 protein or the intracellular free fatty acids content. This highlights that the role of FABP4 in the myocardium in coronary atherosclerosis remains unchanged.

The anatomical and functional vicinity of the perivascular adipose tissue and myocardium puts forward a question about the relationship of the particular components of the lipolytic system between the above-mentioned tissues. However, changes in the mRNA expression of FABP4 and in the protein expression of ATGL only occurred in the myocardium but not in the perivascular adipose tissue. It would suggest that the latter tissue does not influence the behavior of the compounds in the myocardium in coronary artery disease.

## 4. Materials and Methods

### 4.1. Subjects

This investigation was approved by the Bioethics Committee of the Medical University of Bialystok. The study protocol conforms to the ethical guidelines of the 1975 Declaration of Helsinki. All study patients provided written informed consent prior to the participation in the study prior to admission to the Department of Cardiology. The study group consisted of 42 patients with multivessel coronary artery disease confirmed angiographically and qualified for coronary artery bypass grafting (qualification according to the recommendation of the European Society of Cardiology, 2018; designation of the group: CAD). The control group consisted of 11 patients qualified for mitral or aortic valve replacement, but they had no angiographically confirmed coronary artery disease (qualification according to the guidelines of the European Society of Cardiology/European Association of Cardiovascular Imaging, 2017; designation of the group: NCAD). The characteristics of both groups are shown in Table 1. None of the subjects was diabetic and, therefore, did not receive an antidiabetic treatment. It allowed the avoidance of diabetes-induced changes in the function of, e.g., some ion channels involved in crosstalk between the metabolism of myocardial energy substrates and coronary blood flow and thus made the groups more uniform [45,46]. The samples of the right atrial appendage were obtained at the time of the right coronary artery cannulation during elective coronary bypass graft surgery or during the valve’s replacement. The perivascular adipose tissue samples were taken from around of the right coronary artery in close proximity of its aperture. The samples were blotted dry, immediately frozen in liquid nitrogen, and stored at −800 °C until further analyses.

### 4.2. Quantitative Real-Time PCR

Total RNA was isolated from the human’s right atrial appendage and perivascular adipose tissue using the NucleoSpin RNA Plus Kit according to the manufacturer’s protocol (Macherey Nagel GmbH & Co.KG, Duren, Germany). Following RNA purification, DNase treatment (Ambion, Thermo Fisher Scientific, Waltham, MA, USA) was done. The quality of extracted RNA was assessed by spectrophotometry and verified by running the agarose electrophoresis with ethidium bromide. cDNA was generated using iScript cDNA Synthesis Kit (Bio-Rad, Hercules, CA, USA), while specific primers were designed using the Beacon Designer Software (Premier Biosoft, Palo Alto, CA, USA, Table 5 shows primers sequences). Real-time PCR was performed on a Bio-Rad Chromo4 system using SYBR Green JumpStart Taq ReadyMix (Sigma-Aldrich) as the detection dye. Cycling conditions were: 15 s denaturation at 94 °C, 30 s annealing at 60 °C for GAPDH, β-HAD, CS, COX4/1, FAT/CD36, GPAT1, LPL, 61 °C for ATGL, HSL, G0S2 and 62 °C for CGI-58 and 30 s extension at 72 °C for 45 cycles. Melting curve analysis was performed at the end of each reaction to verify PCR product specificity. The mRNA levels of target genes were normalized to human GAPDH and calculated according to the Pfaffl method [47]. All samples were assayed in duplicate.

### 4.3. Protein Extraction and Western Blot

Routine Western blotting procedure was used to examine protein expression (i.e., ATGL, CGI-58, G0S2, HSL, FABP4, β-HAD, CS, COX4/1, FAT/CD36, FAS, SREBP-1c, GPAT1, COX-2, NFkB, 15-LO, and GAPDH), as it was described previously [48,49]. Briefly, the right atrial appendage and perivascular adipose tissue were homogenized in an ice-cold RIPA (radioimmunoprecipitation assay) buffer containing a cocktail of protease and phosphatase inhibitors (Roche Diagnostics GmbH, Mannheim, Germany). The total protein concentration was determined using the bicinchoninic acid (BCA) method with bovine serum albumin (BSA) as a standard. Then, the proteins (20 μg of the total protein) were separated by 10% sodium dodecyl sulfate-polyacrylamide gel electrophoresis and wet transferred onto nitrocellulose membranes. The membranes were incubated overnight at 4 °C with corresponding primary antibodies in a dilution of 1:500. The primary antibodies were purchased from Abcam (i.e., ATGL, HSL, G0S2, FABP4, COX4/1), Novus Biologicals (CGI-58, FAT/CD36), Santa Cruz Biotechnology (GAPDH, β-HAD, CS, COX-2, 15-LO, SREBP-1c), Cell Signaling (FAS, NFkB) and Thermo Fisher Scientific (GPAT1). Thereafter, to detect proteins, anti-rabbit and anti-goat IgG horseradish peroxidase-conjugate secondary antibodies (1:3000; Santa Cruz Biotechnology, Dallas, TX, USA) were used. Equal protein concentration loading was confirmed by routine Ponceau S staining. The protein bands were quantified densitometrically using a ChemiDoc visualization system (Bio-Rad, Poland). Eventually, the protein expression (Optical Density Arbitrary Units) was normalized to GAPDH expression and was related to the control group (patients qualified for mitral or aortic valve replacement but without atherosclerosis).

### 4.4. The Myocardial Lipids

Myocardial triacylglycerols (TG), diacylglycerols (DG), and free fatty acids (FFA) were quantified as described elsewhere [50]. Briefly, the muscle samples were pulverized in an aluminum mortar precooled in liquid nitrogen. The samples were extracted with chloroform-methanol (2:1 vol/vol) with antioxidants (0.01% butylated hydroxytoluene). Moreover, an internal standard (100 μL) containing heptadecanoic acid (C17:0 FFA), 1,2-diheptadecanoin (C17:0 DAG), and triheptadecanoin (C17:0 TAG) (all acids were purchased from Sigma-Aldrich, St., Louis, MO, USA) was added. Lipids were separated into different fractions using thin-layer chromatography (TLC) on silica gel plates (Silica Plate 60, 0.25 mm; Merck, Darmstadt, Germany) with a solvent containing heptane, isopropyl, and acetic acid (60:40:3, vol/vol/vol). Lipid bands were visualized and bands containing the fractions of TG, DG, and FFA were scraped off and methylated. The fatty acid methyl esters (FAMEs) were extracted using pentane. Thereafter samples were dissolved in hexane and analyzed using a Hewlett-Packard 5890 Series II gas chromatograph with Varian CP-SIL88 capillary column (50 mm × 0.25 mm internal diameter) and a flame-ionization detector (FID) (Agilent Technologies, CA, USA). Total TG, DG, and FFA content were estimated as a sum of the individual fatty acids in each fraction. The value was expressed as nanomoles per gram of myocardium wet weight.

### 4.5. Statistical Analysis

The analyses were conducted using Statistica 13.1 (StatSoft, Cracow, Poland). Analysis of variance (ANOVA) followed by Student’s t-test was carried out to determine the existence of differences between the studied groups. If the assumptions of the above tests did not hold, a Kruskal–Wallis rank test with a posthoc pairwise Mann-Whitney U test was applied. The choice of an appropriate method was made upon fulfilling the normality and homogeneity of variances assumptions, and in the case of the violation at least one of the conditions, a non-parametric approach was employed. The normality of the distribution was checked with the Shapiro-Wilk test, and the homogeneity of variances was checked with Levene’s test. The results were presented as mean ± SEM. The statistical significance level was set at 0.05. For consistency of the data, all graphs were box plot (the majority of the data were analyzed by means of non-parametric test). We investigated the codependence between ATGL expression and TG content in human myocardium for both groups. After checking the normality of distributions, Pearson’s correlation coefficients were obtained together with their corresponding p values. Every *p*-value below 0.05 was considered to be statistically significant.

## 5. Conclusions

In conclusion, we are the first to report that the principal components of the triacylglycerol-lipolytic complex are present (at mRNA and protein level) in the human myocardium and in the perivascular adipose tissue in patients with CAD. We found that coronary atherosclerosis reduces the ATGL protein content and induces the accumulation of triacylglycerols in the myocardium. Therefore, herein, we confirmed that ATGL is a principal TG hydrolyzing lipase in the heart and coronary atherosclerosis may be associated with the alteration in ATGL abundance. In the perivascular adipose tissue of the CAD group, the mRNA expression of G0S2, and protein expression of FABP4 were elevated in relation to the respective values in the NCAD group. The latter suggests the increased pro-atherosclerotic potential of the tissue in relation to the coronary vessels.

## Figures and Tables

**Figure 1 ijms-21-00737-f001:**
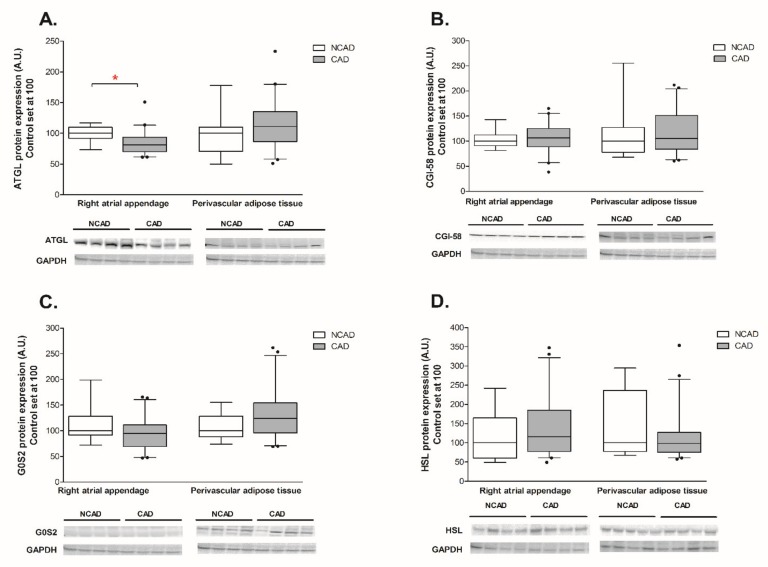
Myocardial and perivascular adipose tissue proteins expression of (**A**) adipose triglyceride lipase (ATGL), (**B**) comparative gene identification 58 (CGI-58), (**C**) G0/G1 switch gene 2 (G0S2), (**D**) hormone-sensitive lipase (HSL) in the coronary atherosclerosis (CAD, *n* = 42) and control (NCAD, *n* = 11) patients. Representative bands of WB analysis were shown. The inner horizontal line of a box represents the median. Box boundaries: 25–75 percentile; whiskers 5–95 percentile. Data are expressed as median ± SEM. For the sake of clarity, the control group median was set at 100, and the CAD group was scaled with respect to NCAD. * *p* < 0.05 vs. control subjects.

**Figure 2 ijms-21-00737-f002:**
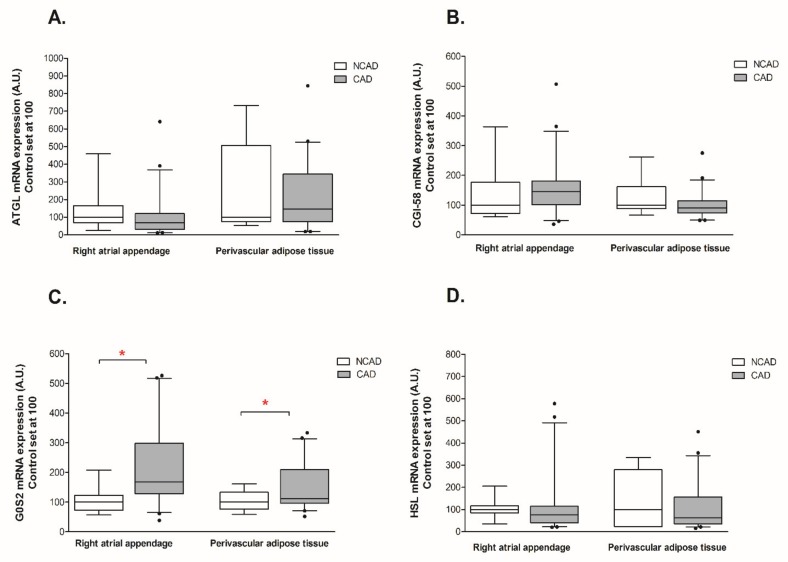
Myocardial and perivascular adipose tissue genes expression of (**A**) adipose triglyceride lipase (ATGL), (**B**) comparative gene identification 58 (CGI-58), (**C**) G0/G1 switch gene 2 (G0S2), (**D**) hormone-sensitive lipase (HSL) in the coronary atherosclerosis (CAD, *n* = 42) and control (NCAD, *n* = 11) patients. The inner horizontal line of a box represents the median. Box boundaries: 25–75 percentile; whiskers 5–95 percentile. Data are expressed as median ± SEM. For the sake of clarity, the control group median was set at 100, and the CAD group was scaled with respect to NCAD. * *p* <0.05 vs. control subjects.

**Figure 3 ijms-21-00737-f003:**
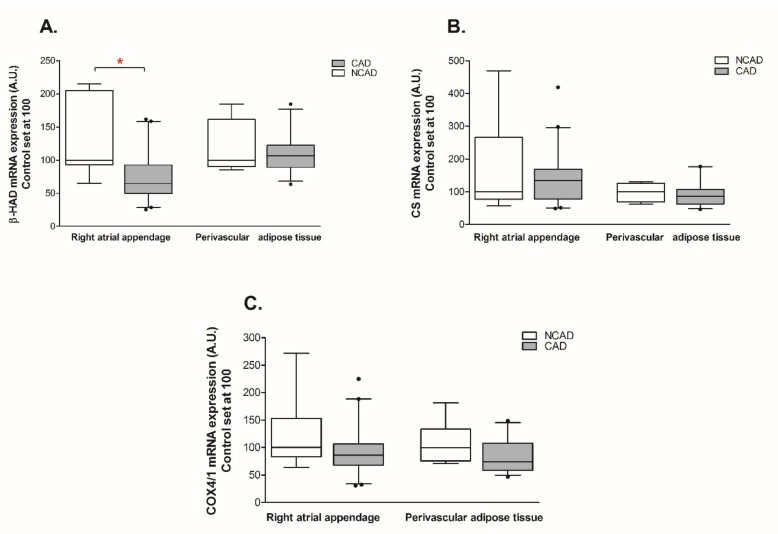
Myocardial and perivascular adipose tissue genes expression of (**A**) beta-hydroxyacyl CoA dehydrogenase (β-HAD), (**B**) citrate synthase (CS), (**C**) cytochrome c oxidase subunit IV isoform 1 (COX4/1) in the coronary atherosclerosis (CAD, *n* = 42) and control (NCAD, *n* = 11) patients. The inner horizontal line of a box represents the median. Box boundaries: 25–75 percentile; whiskers 5–95 percentile. Data are expressed as median ± SEM. For the sake of clarity, the control group median was set at 100, and the CAD group was scaled with respect to NCAD. * *p* < 0.05 vs. control subjects.

**Figure 4 ijms-21-00737-f004:**
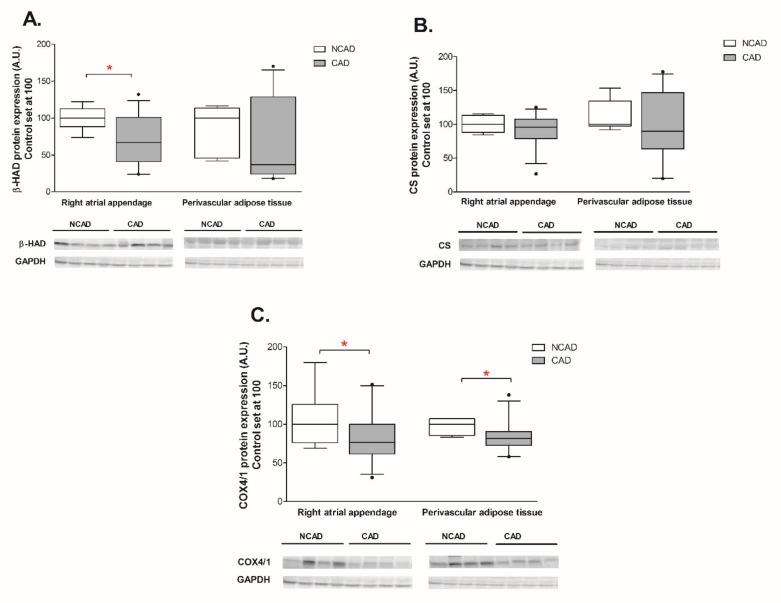
Myocardial and perivascular adipose tissue proteins expression of (**A**) beta-hydroxyacyl CoA dehydrogenase (β-HAD), (**B**) citrate synthase (CS), (**C**) cytochrome c oxidase subunit IV isoform 1 (COX4/1) in the coronary atherosclerosis (CAD, *n* = 42) and control (NCAD, *n* = 11) patients. Representative bands of WB analysis were shown. The inner horizontal line of a box represents the median. Box boundaries: 25–75 percentile; whiskers 5–95 percentile. Data are expressed as median ± SEM. For the sake of clarity, the control group median was set at 100, and the CAD group was scaled with respect to NCAD. * *p* < 0.05 vs. control subjects.

**Figure 5 ijms-21-00737-f005:**
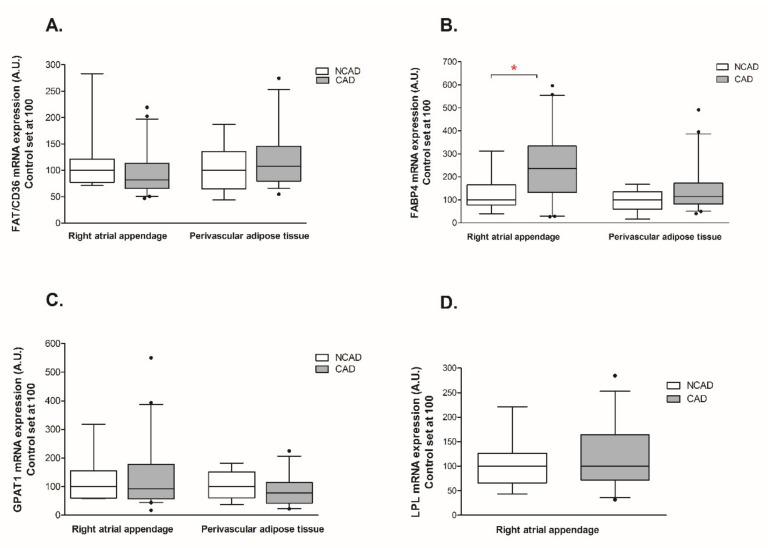
Myocardial and perivascular adipose tissue genes expression of (**A**) fatty acid translocase (FAT/CD36), (**B**) fatty acid-binding protein 4 (FABP-4), (**C**) glycerol-3-phosphate acyltransferase 1 (GPAT1), (**D**) lipoprotein lipase (LPL) in the coronary atherosclerosis (CAD, *n* = 42) and control (NCAD, *n* = 11) patients. The inner horizontal line of a box represents the median. Box boundaries: 25–75 percentile; whiskers 5–95 percentile. Data are expressed as median ± SEM. For the sake of clarity, the control group median was set at 100, and the CAD group was scaled with respect to NCAD. * *p* < 0.05 vs. control subjects.

**Figure 6 ijms-21-00737-f006:**
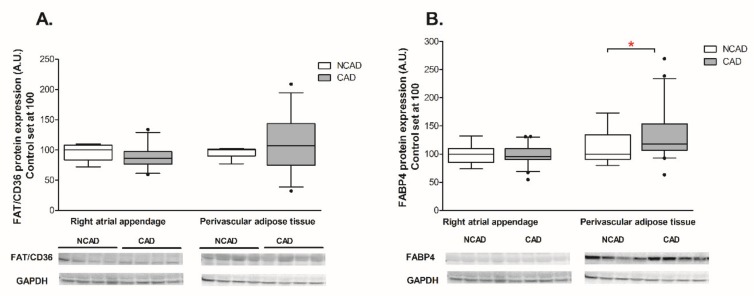
Myocardial and perivascular adipose tissue proteins expression of (**A**) fatty acid translocase (FAT/CD36), (**B**) fatty acid-binding protein 4 (FABP-4) in the coronary atherosclerosis (CAD, *n* = 42) and control (NCAD, *n* = 11) patients. Representative bands of WB analysis were shown. The inner horizontal line of a box represents the median. Box boundaries: 25–75 percentile; whiskers 5–95 percentile. Data are expressed as median ± SEM. For the sake of clarity, the control group median was set at 100, and the CAD group was scaled with respect to NCAD. * *p* < 0.05 vs. control subjects.

**Figure 7 ijms-21-00737-f007:**
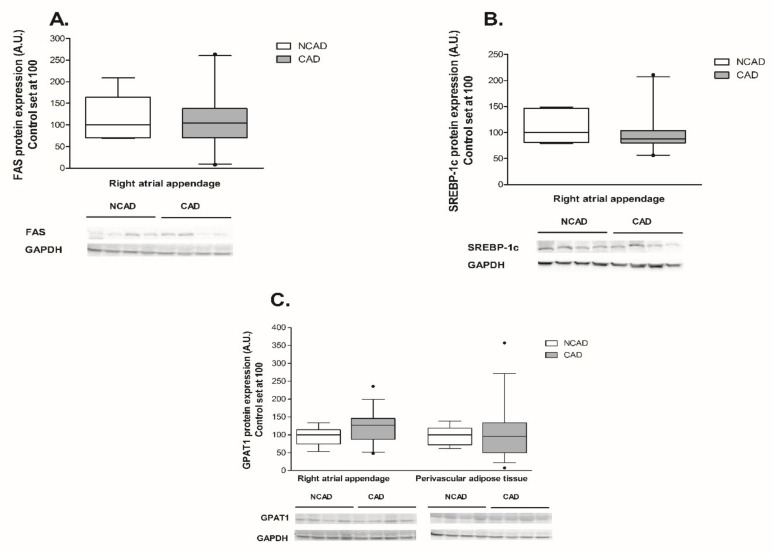
Myocardial proteins expression of (**A**) fatty acid synthase (FAS), (**B**) sterol-regulatory-element-binding protein-1 (SREBP-1c) and perivascular of (**C**) glycerol-3-phosphate acyltransferase 1 (GPAT1) in the coronary atherosclerosis (CAD, *n* = 42) and control (NCAD, *n* = 11) patients. Representative bands of WB analysis were shown. The inner horizontal line of a box represents the median. Box boundaries: 25–75 percentile; whiskers 5–95 percentile. Data are expressed as median ± SEM. For the sake of clarity, the control group median was set at 100, and the CAD group was scaled with respect to NCAD. * *p* < 0.05 vs. control subjects.

**Figure 8 ijms-21-00737-f008:**
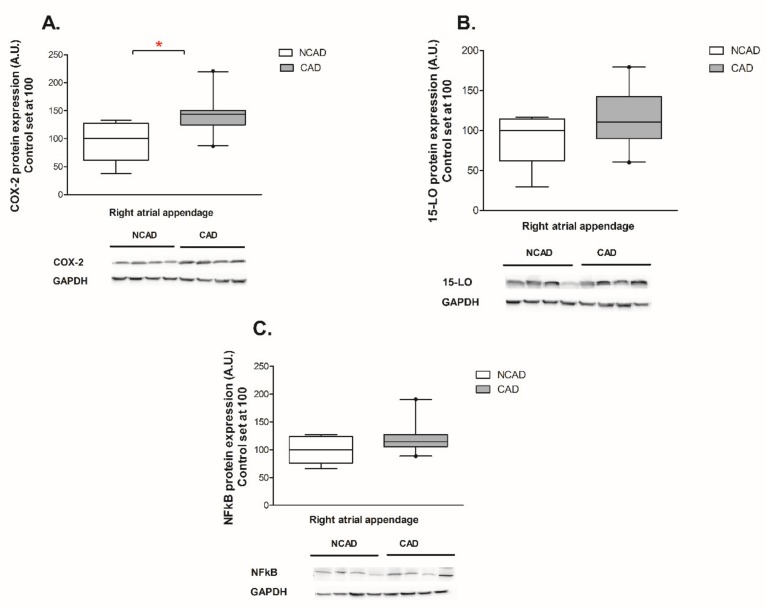
Myocardial proteins expression of (**A**) Cyclooxygenase-2 (COX-2), (**B**) lipoxygenase (15-LO), (**C**) nuclear factor kappa B (NFκβ) in the coronary atherosclerosis (CAD, *n* = 24) and control (NCAD, *n* = 11) patients. The number of subjects with atherosclerosis studied was smaller than the number assayed previous determinations due to an insufficient amount of the muscle tissue available. Representative bands of WB analysis were shown. The inner horizontal line of a box represents the median. Box boundaries: 25–75 percentile; whiskers 5–95 percentile. Data are expressed as median ± SEM. For the sake of clarity, the control group median was set at 100, and the CAD group was scaled with respect to NCAD. * *p* < 0.05 vs. control subjects.

**Figure 9 ijms-21-00737-f009:**
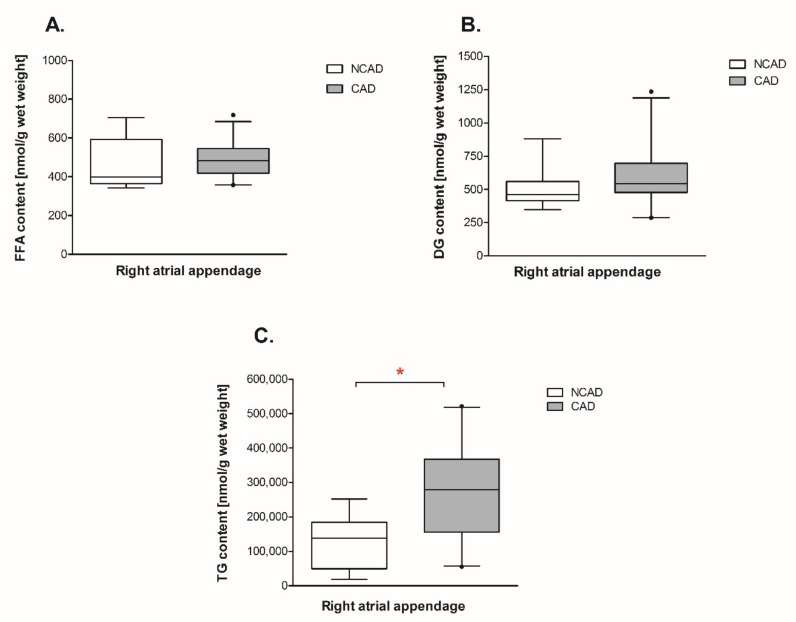
Myocardial content of (**A**) free fatty acids (FFA), (**B**) diacylglycerols (DG) and (**C**) triacylglycerols (TG) in the coronary atherosclerosis (CAD, *n* = 24) and control (NCAD, *n* = 11) patients. The number of subjects with atherosclerosis studied was smaller than the number assayed for mRNA and protein expression due to the insufficient amount of the muscle tissue available. Data are expressed as mean ± SEM. * *p* < 0.05 vs. control subjects.

**Figure 10 ijms-21-00737-f010:**
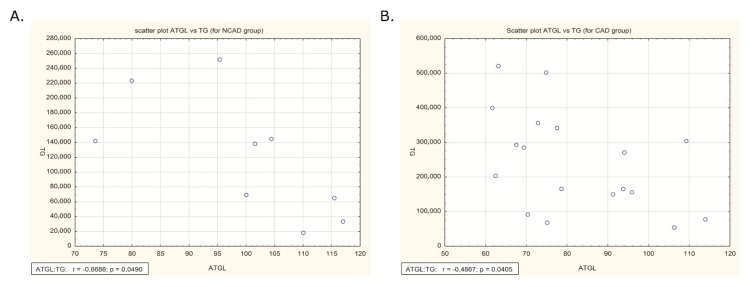
Correlations of ATGL expression and TG content in human myocardium of (**A**) NCAD and (**B**) CAD patients.

**Table 1 ijms-21-00737-t001:** Clinical characteristics of the subjects. Data are presented as mean ± SD.

Item	Control (NCAD) (*n* = 11)	Coronary Atherosclerosis (CAD) (*n* = 42)	*p* Value
Age (years)	61.1 ± 8.3	63.6 ± 8.3	0.37
Sex (M/F)	5/6 (45.4/55.6%)	40/2 (95.3/4.7%) *	0.008
BMI (kg/m^2^)	26.7 ± 5.2	27.3 ± 3.2	0.68
Hypertension ^#^	5 (45.4%)	34 (80.9%) *	0.02
Hyperlipidemia	8 (72.7%)	32 (76.1%)	1.0
History of infarction	0	17 (40.4%) *	0.012
Left ventricular ejection fraction (%)	52.2 ± 6.8	53.1 ± 9.5	0.76
Fasting blood glucose (mg/dL)	95.5 ± 10.8	96.2 ± 14.1	0.94
Creatinine clearance (ml/min)	86.2 ± 23.6	91.9 ± 0.2	0.37
Total cholesterol (mg/dL)	198.1 ± 30.2	183.4 ± 41.7	0.32
HDL cholesterol (mg/dL)	53.9 ± 10.5	46.6 ± 11.3	0.065
LDL cholesterol (mg/dL)	120.4 ± 17.9	109.9 ± 36.6	0.45
Triglyceride (mg/dL)	112.4 ± 22.5	152.9 ± 83.7	0.14
C-reactive protein (mg/L)	2.0 ± 2.0	4.9 ± 9.5	0.25
Hemoglobin (g/dL)	13.3 ± 1.1	13.7 ± 1.4	0.42
Platelets (×10^3^/mm^3^)	195.8 ± 29.9	232.8 ± 49.6 *	0.03
HbA1C (%)	5.5 ± 1.1	5.9 ± 0.3	0.43
Alanine aminotransferase (U/L)	25.1 ± 12.7	52.9 ± 42.6 *	0.02
Aspartate aminotransferase (U/L)	30.6 ± 15.7	39.9 ± 28.4	0.27
**Medication:**			
ACEI/ ARB	4 (36.4%)	40 (95.2%) *	0.00001
Beta-blockers	10 (90.9%)	41 (97.6%)	0.78
Statins	6 (54.5%)	41 (97.6%) *	0.000001
Proton pump inhibitors	10 (90.9%)	39 (92.8%)	0.83
Spironolaktone	3 (27.3%)	7 (16.6%)	0.42
Furosemide	4 (36.4%)	8 (19.0%)	0.22
Aspirin	None	42 (100%)	0.00

* *p* <0.05 for different patients with multivessel coronary artery disease (coronary artery bypass grafting) and control patients with no atherosclerosis (mitral or aortic valve replacement). ACEI/ARB—angiotensin-converting enzyme inhibitor/angiotensin receptor blocker. ^#^ According to the guidelines of the European Society of Cardiology/European Society of Hypertension 2018.

**Table 2 ijms-21-00737-t002:** TG—Fatty acid composition (nmol/g of wet tissue).

Composition	NCAD	CAD
Myristic acid (14:0)	4282.14 ± 2613.1 (3.66%)	9158.87 ± 5719.6 (3.33%) *
Palmitic acid (16:0)	38,589.25 ± 25,242.1 (31.9%)	84,158.08 ± 45,434.1 (31.31%) *
Palmitooleic acid (16:1)	10,635.3 ± 7981.3 (8.48%)	27,838.65 ± 16,791.5 (10.22%) *
Stearic acid (18:0)	5371.43 ± 2945.7 (4.95%)	9332.44 ± 5345.3 (3.64%) *
Oleic acid (18:1n9c)	48,556.93 ± 35,024.1 (39.48%)	109,021.54 ± 57,447.9 (40.86%) *
Linoleic acid (18:2n6c)	10,304.8 ± 8165.3 (8.69%)	20,929.94 ± 11,997.4 (7.93%) *
Arachidic acid (20:0)	70.65 ± 35.7 (0.08%)	122.86 ± 72.2 (0.05%) *
Linolenic acid (C18:9n3)	1855.38 ± 1434.9 (1.6%)	3744.42 ± 2893.9 (1.4%)
Behenic acid (22:0)	34.93 ± 18.2 (0.04%)	110.63 ± 57.7 (0.04%) *
Arachidonic acid (20:4n6)	634.53 ± 429.7 (0.59%)	1469.99 ± 767 (0.58%) *
Eicosapentaenoic acid (20:5n3)	17.01 ± 11.9 (0.02%)	116.3 ± 100.7 (0.04%) *
Nervonic acid (24:1)	16.52 ± 9.8 (0.02%)	25.71 ± 14.2 (0.01%)
Docosahexaenoic acid (22:6n3)	498.41 ± 353.5 (0.42%)	1292.91 ± 805.5 (0.5%) *
Lignoceric acid (24:0)	104.82 ± 71.7 (0.09%)	272.75 ± 145.9 (0.11%) *
Saturated	48,453.23 ± 30,585.3 (40.71%)	103,155.63 ± 56,045.7 (38.47%) *
Unsaturated	72,518.89 ± 51,264.1 (59.29%)	164,439.46 ± 87,040.3 (61.53%) *
Total	120,972.12 ± 81,391.2 (100%)	267,595.09 ± 140,943.4 (100%) *

Effects of coronary atherosclerosis on myocardial triacylglycerol content and FA composition (mean ± SEM, *n* = 11 per NCAD group, *n* = 24 per CAD group). ∗ Difference versus NCAD group (*p* <0.05). Values are expressed in nmol/g of wet tissue and show the percentage share of the FA in relation to the total TG content.

**Table 3 ijms-21-00737-t003:** DG—Fatty acid composition (nmol/g of wet tissue).

Composition	NCAD	CAD
Myristic acid (14:0)	47.98 ± 8.3 (9.9%)	58.16 ± 25.1 (9.4%)
Palmitic acid (16:0)	184.72 ± 59.6 (36.3%)	232.98 ± 97.1 (37.4%)
Palmitooleic acid (16:1)	24.18 ± 12.3 (4.6%)	40.08 ± 27.1 (6%)
Stearic acid (18:0)	96.22 ± 18.5 (19.8%)	85.84 ± 19 (15%)
Oleic acid (18:1n9c)	81.82 ± 41.3 (15.5%)	118.47 ± 61.2 (18.5%)
Linoleic acid (18:2n6c)	29.97 ± 16 (5.7%)	39.83 ± 17 (6.4%) *
Arachidic acid (20:0)	1.74 ± 0.4 (0.4%)	2.02 ± 1 (0.3%)
Linolenic acid (C18:9n3)	4.67 ± 2.1 (0.9%)	4.71 ± 3 (0.7%)
Behenic acid (22:0)	1.28 ± 0.5 (0.3%)	1.38 ± 0.4 (0.2%)
Arachidonic acid (20:4n6)	26.26 ± 13.2 (5.1%)	22.09 ± 6.5 (4.1%)
Eicosapentaenoic acid (20:5n3)	1.94 ± 0.8 (0.4%)	1.86 ± 1 (0.3%)
Nervonic acid (24:1)	0.82 ± 0.3 (0.2%)	0.89 ± 0.3 (0.2%)
Docosahexaenoic acid (22:6n3)	3.69 ± 2.1 (0.7%)	4.47 ± 1.7 (0.8%)
Lignoceric acid (24:0)	1.03 ± 0.7 (0.2%)	2.66 ± 2.3 (0.5%) *
Saturated	332.98 ± 77.1 (66.9%)	383.04 ± 134.5 (62.9%)
Unsaturated	173.34 ± 82.9 (33.1%)	232.41 ± 105.2 (37.1%)
Total	506.32 ± 157.5 (100%)	615.45 ± 234.7 (100%)

Effects of coronary atherosclerosis on myocardial diacylglycerol content and FA composition (mean ± SEM, *n* = 11 per NCAD group, *n* = 24 per CAD group). ∗ Difference versus NCAD group (*p* <0.05). Values are expressed in nmol/g of wet tissue and showed the percentage share of the FA in relation to the total DG content.

**Table 4 ijms-21-00737-t004:** FFA—Fatty acid composition (nmol/g of wet tissue).

Composition	NCAD	CAD
Myristic acid (14:0)	26.68 ± 5.7 (5.9%)	26.38 ± 12 (5.3%)
Palmitic acid (16:0)	167.38 ± 42.2 (36.3%)	173.44 ± 30.1 (35.9%)
Palmitooleic acid (16:1)	13.63 ± 5.9 (2.9%)	15.16 ± 5.9 (3.1%)
Stearic acid (18:0)	103.15 ± 19.5 (22.9%)	102.87 ± 18.5 (21.6%)
Oleic acid (18:1n9c)	74.79 ± 35.1 (15.4%)	88.02 ± 17.5 (18.3%)
Linoleic acid (18:2n6c)	32.77 ± 15.5 (6.7%)	31.88 ± 11.9 (6.5%)
Arachidic acid (20:0)	1.46 ± 0.3 (0.3%)	1.74 ± 0.6 (0.4%)
Linolenic acid (C18:9n3)	5.15 ± 2.5 (1.1%)	3.91 ± 2.4 (0.8%)
Behenic acid (22:0)	0.99 ± 0.3 (0.2%)	1.14 ± 0.5 (0.2%)
Arachidonic acid (20:4n6)	31.65 ± 19.2 (6.8%)	29.96 ± 19.6 (6.1%)
Eicosapentaenoic acid (20:5n3)	0.93 ± 0.2 (0.2%)	1.63 ± 1 (0.3%) *
Nervonic acid (24:1)	0.59 ± 0.2 (0.1%)	0.68 ± 0.4 (0.1%)
Docosahexaenoic acid (22:6n3)	3.93 ± 3.3 (0.8%)	2.55 ± 1.2 (0.5%)
Lignoceric acid (24:0)	1.5 ± 0.6 (0.3%)	3.16 ± 3 (0.7%)
Saturated	301.17 ± 60.8 (66%)	308.72 ± 47 (64.1%)
Unsaturated	163.43 ± 70.6 (34%)	173.8 ± 36.9 (35.9%)
Total	463.31 ± 128.8 (100%)	482.52 ± 75.9 (100%)

Effects of coronary atherosclerosis on myocardial free fatty acids content and FA composition (mean ± SEM, *n* = 11 per NCAD group, *n* = 24 per CAD group). ∗ Difference versus NCAD group (*p* <0.05). Values are expressed in nmol/g of wet tissue and showed the percentage share of the FA in relation to the total FFA content.

**Table 5 ijms-21-00737-t005:** Primers sequences for real-time PCR analysis.

Gene	Primer Sequence	
	Forward	Reverse
*GAPDH*	5′-AAGCCTGCCGGTGACTAAC-3′	5′-GTTAAAAGCAGCCCTGGTGAC-3′
*ATGL*	5′-GCTTCCTCGGCGTCTACTAC-3′	5′-CAATGAACTTGGCACCAGCC-3′
*G0S2*	5′-ACCACAAGCATCCACCAA-3′	5′-GCATTTATCCTTCCTCCCTA-3′
*CGI-58*	5′-AGACCCAGGTTTGACAGTGATG-3′	5′-AGTAAGCAGCAGCCAAGAATCC-3′
*HSL*	5′-CACGATGGGTGGAATGGTGG-3′	5′-ACCAGCGACTGTGTCATTGT-3′
*FABP4*	5′-GGGCCAGGAATTTGACGAAG-3′	5′-AACTCTCGTGGAAGTGACGC-3′
*FAT/CD36*	5′-AAGTCACTGCGACATGATTAATGG-3′	5′-GAACTGCAATACCTGGCTTTTCTC-3′
*GPAT1*	5′-AACCCCAGTATCCCGTCTTT-3′	5′-CAGTCACATTGGTGGCAAAC-3′
*β-HAD*	5′-CTTGCTCCGAGAGGGAGTC-3′	5′-AGCTCGTAGCTGGGAGGAAC-3′
*CS*	5′-GATTGTGCCCAATGTCCTCT-3′	5′-TTCATCTCCGTCATGCCATA-3′
*COX4/1*	5′-GGTCACGCCGATCCATATAAG-3′	5′-TCTGTGTGTGTACGAGCTCATGA-3′
*LPL*	5′–GAGATTTCTCTGTATGGCACC-3′	5′–CTGCAAATGAGACACTTTCTC-3′

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
