# Peer review of "The Gene and Protein Expression of the Main Components of the Lipolytic System in Human Myocardium and Heart Perivascular Adipose Tissue. Effect of Coronary Atherosclerosis"

_ijms, 2020, doi:10.3390/ijms21030737_

Round 1
Reviewer 1 Report
I read with great interest the present article on the presence of the principal components of the triacylglycerol lipolytic complex in myocardium and perivascular adipose tissue. Authors observed that severe coronary atherosclerosis affects both the mRNA and protein expression the chief components of some complex. One of the most remarkable findings of this study is the reduction in the ATGL protein expression in human myocardium of the CAD patients. The results are appealing. I only have a few minor revisions to do:
1) In the population description, 42 patients with multivessel coronary artery disease qualified to coronary artery bypass grafting. I would suggest referring to any specific guidelines for the recommendations.
2) In the population description, 11 patients without atherosclerosis, qualified to mitral or aortic valve replacement, I would suggest referring to any specific guidelines for the recommendations.
3) In CAD group 34 patients (80.9 %) had hypertension and 17 (40.4%) had a history of infraction. Please correct INFRACTION with “myocardial infarction”. I would also suggest clarifying definition of hypertension.
4) All the subjects participated in the study were without diabetes mellitus. Is there any specific reason? I would say to clarify it at this point. In the results Table, HbA1C in the CAD group was 5,9 ± 0,3 (max 6,2%) that is were close to 6,5% (cut-off for diabetes diagnosis). Is this a result of any therapy? This is a crucial point. It is well known that an elevation of TG has also been demonstrated in diabetes type 2, and it is well correlated with oxidative stress and ischemic heart disease by a complex pathway involving some ion channels acting in the cross-talk between myocardial energy state and coronary blood flow (refer to J Diabetes Res. 2019 Apr 4;2019:9489826 and Int J Mol Sci. 2018 Mar 10;19(3). pii: E802). Please clarify this concept.
5) Although some patient has a history of myocardial infarction, there is no mention on antiplatelet therapy. Please give more details
6) In the CAD group LDL and total cholesterol were lower, probably due to statin use. It is well known that statin (simvastatin, atorvastatin, rosuvastatin) lower triglycerides, however not in the CAD group (compared with NCAD). Is there any explanation?
Author Response
I read with great interest the present article on the presence of the principal components of the triacylglycerol lipolytic complex in myocardium and perivascular adipose tissue. Authors observed that severe coronary atherosclerosis affects both the mRNA and protein expression the chief components of some complex. One of the most remarkable findings of this study is the reduction in the ATGL protein expression in human myocardium of the CAD patients. The results are appealing. I only have a few minor revisions to do:
Dear Reviewer 1,
Thank You for an in-depth analysis of our study and constructive comments. We appreciate your work and time spent on reviewing the manuscript. Below You will find our answers to the discussed issues. For Your greater convenience we have placed them (as well as all the changes in the manuscript) with red font. We hope that You will find them satisfactory.
Best regards, Agnieszka Miklosz – corresponding author
1) In the population description, 42 patients with multivessel coronary artery disease qualified to coronary artery bypass grafting. I would suggest referring to any specific guidelines for the recommendations.
The following is included in the Materials and Methods of the manuscript, section 4.1 “Subjects”: (qualification according to the recommendation of the European Society of Cardiology, 2018).
2) In the population description, 11 patients without atherosclerosis, qualified to mitral or aortic valve replacement, I would suggest referring to any specific guidelines for the recommendations.
The following was included in the Materials and Methods of the manuscript, section 4.1 “Subjects”: (qualification according to the guidelines of European Society of Cardiology/European Association of Cardiovascular Imaging, 2017
3) In CAD group 34 patients (80.9 %) had hypertension and 17 (40.4%) had a history of infraction. Please correct INFRACTION with “myocardial infarction”. I would also suggest clarifying definition of hypertension.
“Myocardial” was added. The following was added in the foot note of the Table 1: # according to the guidelines of the European Society of Cardiology/European Society of Hypertension 2018.
4) All the subjects participated in the study were without diabetes mellitus. Is there any specific reason? I would say to clarify it at this point. In the results Table, HbA1C in the CAD group was 5,9 ± 0,3 (max 6,2%) that is were close to 6,5% (cut-off for diabetes diagnosis). Is this a result of any therapy? This is a crucial point. It is well known that an elevation of TG has also been demonstrated in diabetes type 2, and it is well correlated with oxidative stress and ischemic heart disease by a complex pathway involving some ion channels acting in the cross-talk between myocardial energy state and coronary blood flow (refer to J Diabetes Res. 2019 Apr 4;2019:9489826 and Int J Mol Sci. 2018 Mar 10;19(3). pii: E802). Please clarify this concept.
The following was added in the section of “Subjects”: None of the subjects was diabetic and therefore did not received an antidiabetic treatment. It allowed to avoid changes in the function of, eg. some ion channels involved in crosstalk between metabolism of myocardial energy substrates and coronary blood flow and thus made the groups more uniform. The recommended references are included in the materials and methods section.
5) Although some patient has a history of myocardial infarction, there is no mention on antiplatelet therapy. Please give more details
”Aspirin” was added in the table one.
6) In the CAD group LDL and total cholesterol were lower, probably due to statin use. It is well known that statin (simvastatin, atorvastatin, rosuvastatin) lower triglycerides, however not in the CAD group (compared with NCAD). Is there any explanation?
It is hard to say. It is presumed that the therapeutic aim in CAD patients was to lower the plasma LDL level. A physician likely tried to meet the goal and did not pay enough attention to the plasma triglyceride concentration. We did not include this presumption in the manuscript.
Reviewer 2 Report
The author demonstrated the triacylglycerol lipolytic gene change in myocardium and perivascular adipose tissue in patients with coronary artery disease and suggested severe coronary atherosclerosis affects triacylglycerol lipolytic complex including reduction of ATGL expression. The finding is extremely interesting and very clinical relevant. There is no doubt that this information is definitely worth publishing and revealed.
Thought the downside of the current work is that the provided evidence is very descriptive and lack of causal relationship; plus, the proposed pathway(triacylglycerol lipolytic gene change) is weak and lacks other strong data to support such that the overall conclusion is weak and shaky.
Comments that are doable to make the current draft better are listed below:
The title is “Coronary atherosclerosis alters gene and protein expressions so of the main ….). I disagree that it is atherosclerosis that causes gene alterations. There is no data included here to support the causal relationship. The author can not rule out the possibility that because of changes of those genes, atherosclerosis develops.
However, if the author still prefers to keep this title, it is possible that the author can reanalyze the data and correlate the disease severity with gene alterations of myocardium or perivascular adipose tissue to get clues. I think it will be informative to try.
There is a major concern that most of patients included here are male (95%) compared to controls ( 45%). Since it is well known that there is a gender effect in cardiovascular disease between male and female, it is possible this improper control may lead to the totally wrong conclusion; after all, the overall phenotype is not drastic (for example, fig2A with only 20% difference). How could 20% enzymatic activity difference make 2 fold TG difference? The author should make the male percentage in controls even compared to the patient group. Or the author should separate male versus female of the control and the patient group and show there is no gender effect, at least in controls. The number of the control in this study even without separation of male and female is low compared to other clinical studies.
Since up to 95% of patients take statins or other drugs, is that possible the increased TG levels is due to the drug effect coming from statins? It is known that statins cause insulin insensitivity; insulin signaling is highly relevant to lipid handling in cells. Since the overall phenotype of ATGL change is fairly small (Fig2A, 20%), the increased TG (2 fold) in tissues may not exclusively come from the reduced ATGL; instead, it may come from insulin signaling that leads to de novo fatty acid synthesis. To prove it, the author could measure plasma insulin in controls versus patients. Meanwhile, the author should also test mRNA/protein levels of fatty acid synthesis pathway, such as SREBP1c, FAS, ACC, FASN, etc…
It will be also helpful to perform western blot of AKT, mTOR as readouts for the insulin signaling.
In addition to fatty acid synthesis, is that possible that lipoprotein lipase (LPL) expression is higher in patients such that there is more FFA that feeds to CD36 to transport, although the author showed the CD36 expression between controls versus patients is the same.
https://ahajournals.org/doi/full/10.1161/01.cir.102.14.1629
It will be helpful to test LPL mRNA or protein levels.
If results from points 3 and 4 show no difference, it will justify the importance of the pathway the author proposed here. It will then be important to reanalyze the data to correlate the expression of ATGL and TG levels to show the correlation to make the point valid.
Since TG of myocardial content is increased in patients (Fig7C), should lipid droplets that store TG be increased as well? Can the author measure lipid droplet markers such as perilipin from mRNA or protein levels to support the data ?
If TG of myocardial content is really increased in patients, what is the function of the heart loaded with more TG? Is it more inflammatory? Can the author measure cytokine expression such as TNFa, IL1b, etc…?
In conclusion, the data is very interesting, but still need other evidence to support the finding. In my view, the title of the current work is overinterpreted. There is no causal relationship based on data provided to link atherosclerosis to myocardium or perivascular adipose tissue. If the correlation between disease severity and gene alteration of myocardium and perivascular adipose tissue can not be developed, the title needs to be modified.
In order to at least make the data from the myocardium solid, it is needed to demonstrate why only 20% difference of enzymatic activity of ATGL could lead to 2 fold differences of TG. Controls exps including FFA flux in or lipid de novel synthesis need to be tested.
Minor:
Fig4A, I disagree HAD expression in CAD is lower than NCAD based on the blot.Author Response
Please see the attachment

Round 2
Reviewer 2 Report
After the revision, the quality of current draft is greatly improved.
Here are some comments for the minor revision.
The author replied" How could 20% enzymatic activity difference make 2 fold TG difference? We presume that it a matter of time". Please include the thought in the discussion. The author replied” However, it is very difficult to collect a group of subjects at advanced age, without coronary atherosclerosis, qualified to the heart surgery because of valval diseases. Therefore, we decided to gather both sexes to the control group”. Please include the thought in the discussion. It is understandable that it takes time and efforts to gather both sexes; however, since the mRNA difference is only 20% and the enzyme activity is NOT measured and left for the future examination, the overall conclusion can be totally changed when the number of control samples from both sexes are provided. It will be cautious and rigorous to list this concern/limitation. Similarly, please also mention in the discussion that the phenotype from patients who take different drugs could be also affected by the drug effect, although the current evidence generally supports the conclusion.